# Monitoring Farmed Fish Welfare by Measurement of Cortisol as a Stress Marker in Fish Feces by Liquid Chromatography Coupled with Tandem Mass Spectrometry

**DOI:** 10.3390/molecules27082481

**Published:** 2022-04-12

**Authors:** Vanessa Andrea Meling, Kjetil Berge, David Lausten Knudsen, Per Ola Rønning, Cato Brede

**Affiliations:** 1Department of Mechanical, Electronic and Chemical Engineering, Faculty of Technology, Art and Design, Oslo Metropolitan University, N-0166 Oslo, Norway; vanessa_meling@hotmail.com (V.A.M.); peror@oslomet.no (P.O.R.); 2Skretting AS, N-4016 Stavanger, Norway; kjetil.berge@skretting.com; 3Fishlab AS, N-4015 Stavanger, Norway; david.knudsen@fishlab.no; 4Department of Chemistry, Bioscience and Environmental Engineering, University of Stavanger, N-4036 Stavanger, Norway; 5Department of Medical Biochemistry, Stavanger University Hospital, N-4068 Stavanger, Norway

**Keywords:** aquaculture, fish, salmon, feces, deconjugation, hydrolysis, extraction, chromatography, mass spectrometry, LC-MS/MS

## Abstract

The aquaculture industry has become a sustainable source of food for humans. Remaining challenges include disease issues and ethical concerns for the discomfort and stress of farmed fish. There is a need for reliable biomarkers to monitor welfare in fish, and the stress hormone cortisol has been suggested as a good candidate. This study presents a novel method for measurement of cortisol in fish feces based on enzymatic hydrolysis, liquid–liquid extraction, derivatization, and finally instrumental analysis by liquid chromatography coupled with tandem mass spectrometry. Hydrolysis and extraction conditions were optimized. Cortisol appeared to be mostly conjugated to sulfate and less conjugated to glucuronic acid in the studied samples of feces from farmed Atlantic salmon. The method was suitable for quantification of cortisol after enzymatic deconjugation by either combined glucuronidase and sulfatase activity, or by glucuronidase activity alone. The limit of detection was 0.15 ng/g, the limit of quantification was 0.34 ng/g, and the method was linear (R^2^ > 0.997) up to 380 ng/g, for measurement of cortisol in wet feces. Method repeatability and intermediate precision were acceptable, both with a coefficient of variation (CV) of 11%. Stress level was high in fish released into seawater, and significantly reduced after eight days.

## 1. Introduction

In Norway, aquaculture has grown to become the second-largest export industry, after oil and gas [1], and is now producing more than half of the total volume in Europe [2]. The most common farmed fish is Atlantic salmon (*Salmo salar*). Despite of its success, the industry is still facing issues around fish health and welfare that need to be resolved [3]. With the typical production in surface-based cages, fish health can be challenged by a range of events, including weather, parasites, algae, jellyfish, diseases, reduced oxygen, and crowding by net deformation [4]. Following a jellyfish bloom occurrence for example, both direct and indirect effects on fish health can be observed. Jellyfish are known to cause biofouling of nets, which reduces water flow and leads to accumulation of solids and an oxygen reduction [5]. Furthermore, fish can be exposed to strong currents and waves when moving nets to new locations [6]. Stressful events such as these directly reduce the appetite and growth rate of fish [7] and make them more susceptible to diseases [5]. Therefore, identification of early markers of stress can be of high value, not only to prevent an outbreak of disease and economical losses, but also to ensure proper ethical farming conditions.

The major reaction control system for stress in animals is the hypothalamic–pituitary–adrenal (HPA) axis [8]. An equivalent to this system is the hypothalamic–pituitary–intrarenal (HPI) axis found in teleost fish [9,10]. Triggering of the system during stressful events elevates the production of corticosteroid hormones, resulting in higher levels of cortisol measured in the blood [11,12,13]. It is therefore possible to study stress in fish by monitoring the level of cortisol [14]. Measurement of cortisol in feces is an alternative approach and could potentially correlate with stress experienced some time before the sampling is carried out. Furthermore, collection of feces from fish is a less invasive procedure [15]. Direct measurement of corticoid metabolites in fish feces has previously been achieved by using enzyme-linked immunosorbent assay (ELISA), and this type of assessment was then proposed as a good indicator of fish welfare [16]. It is well known that immunoassays, such as ELISA, can overestimate analyte concentrations, especially when there is cross-reactivity with metabolites. Analysis by high-performance liquid chromatography coupled with UV detection (HPLC-UV) has also been reported [17], but has the disadvantage of low sensitivity. Today, liquid chromatography coupled with tandem mass spectrometry (LC-MS/MS) is well established in clinical laboratories for selective and sensitive measurement of cortisol and other steroid hormones at low concentration levels in biological samples [18]. This technique utilizes electrospray ionization (ESI), where other methods often depend on derivatization to improve ionization efficiency [19]. For this purpose, we recently investigated several derivatization reagents suitable for improved detectability of steroid hormones in saliva [20]. In the present work, we present a novel method for the measurement of cortisol in fish feces by LC-MS/MS. Before instrumental analysis, samples were deconjugated by beta-glucuronidase enzymes, either with or without sulfatase activity. Increased detectability of cortisol was achieved by liquid–liquid extraction and derivatization. Finally, we applied the new method for analysis of feces from farmed Atlantic salmon, allowing us to identify fish groups with elevated stress levels.

## 2. Materials and Methods

### 2.1. Chemicals and Solutions

Cortisol and isotopic-labeled D4 cortisol (hydrocortisone-9,11,12,12-d4), used as internal standard, were both from Merck KGaA (Darmstadt, Germany). LC-MS-grade methanol and acetonitrile (ACN), and analytical-grade 1-propanol, 1-butanol, methyl tert-butyl ether (MTBE), sodium chloride, hydrochloric acid, acetic acid, formic acid, and 25% ammonium hydroxide solution were all from VWR International (Radnor, PA, USA). Water was type I, purified to 18.2 MOhm (MilliporeSigma, Burlington, MA, USA). Aqueous solutions of beta-glucuronidase from *Helix Pomatia* (product number G1707) and *Patella vulgata* (product number G2174) were from Merck. Beta-glucuronidase with sulfatase (Abalonase™+) or without added sulfatase (Abalonase™) were from Ango (San Ramon, CA, USA). 4-Aminobenzoic hydrazide (4-ABH) was from TCI (Tokyo, Japan).

Stock and intermediate solutions of cortisol were made with methanol, while eight calibrator-standard solutions in the 0–100 ng/mL range were made by dilution with water. Stock and user solution of D4 cortisol internal standard (67 ng/mL) were made with methanol. Solutions prepared in methanol are stable at 4 °C for at least 2 years in our experience (Stavanger University Hospital). Saturated sodium chloride (NaCl) solution (approximately 5 mol/L) was made by adding more salt than could be dissolved in water at room temperature. Ammonium acetate buffer (2 mol/L) was made by pipetting 5 mL of glacial acetic acid (18 mol/L) into water and adding either 3 mL or 5 mL ammonium hydroxide solution (13 mol/L) to make up a buffer with pH 5 or 6, respectively, after dilution to a total volume of 45 mL in graduated polypropylene vials. The 4-ABH user solution (10 mg/mL) was made by dissolving 250 mg 4-ABH with 25 mL methanol containing 50 mM HCl.

### 2.2. Samples

A pooled sample of fish feces from Atlantic salmon (Salmo salar) was kindly provided by Skretting AS (Stavanger, Norway) for method development. In addition, Skretting AS also provided 60 samples of feces from individual fish from a health trial conducted under commercial farming conditions in Southern Norway (production site no. 11971, Store Teistholmen, Rogaland, Grieg Seafood ASA (Bergen, Norway)). The fish had a mean average weight of 240 ± 120 g (*n* = 57) and were anesthetized by an overdose with Finquel (MS-222) before sampling. Feces samples were kept frozen (−18 °C) prior to analysis. Individual fish were sampled a few days after being transferred by well boat from a freshwater facility on land to open sea cages in the fjord. A total of 20 fish were sampled 4 days after transfer, and additional 10 fish were sampled on day 5, 6, 7, and 8 after transfer.

### 2.3. Sample Preparation

#### 2.3.1. Fish Feces Supernate

A pooled sample of fish feces was applied for method development and for investigation of analytical performance characteristics. A measure of 2.5 g of pooled fish feces was transferred to a 10 mL polypropylene tube with a screw cap (Sarstedt, Nümbrecht, Germany) followed by addition of 7 mL water, manual shaking for 1 min, and finally centrifugation at 2000× *g* for 5 min. Particle-free pooled supernate from multiple such preparations was transferred to large 50 mL polypropylene tubes (Teknolab, Ski, Norway) and stored frozen until use.

#### 2.3.2. Deconjugation

Deconjugation experiments were performed in triplicates with 500 μL aliquots of the pooled supernate transferred to 2 mL polypropylene (PP) microcentrifuge tubes (Teknolab). Both deconjugation and subsequent liquid–liquid extraction (LLE) were carried out in the same vial. For optimization of the enzymatic hydrolysis, 50 μL ammonium acetate buffer (2 M, pH 5 or 6) was added to the 500 μL supernate samples, followed by 20 μL enzyme solution, and finally incubation at 60 °C for 1 h. Upon identification of beta-glucuronidase from *Helix Pomatia* as the most promising enzyme, further optimization was carried out with using 10, 20, 30, 40, and 50 μL enzyme solution, and also by incubation at 60 °C for 1, 2, 3, and 24 h. Chemical hydrolysis was attempted by adding 20 μL of 2, 4, 6, 8, or 10% (*v*/*v*) of either sulfuric acid or ammonium hydroxide, followed by incubation at 60 °C for 1 h.

After deconjugation, samples were processed by an LLE procedure as follows: 50 μL internal standard user solution, 700 μL MTBE, and 100 μL saturated NaCl were added to all vials, followed by manual shaking for 1 min, and centrifugation for 5 min at 2000× *g*. A measure of 400 μL of the solvent phase was transferred to a new 2 mL PP tube and evaporated to dryness by vacuum centrifugation at 60 °C for 20 min (Eppendorf, Hamburg, Germany). Derivatization was carried out by adding 20 μL 4-ABH user solution, which was incubated at room temperature for 30 min before adding 40 μL water and being transferred to 0.3 mL PP autosampler vials (VWR). Similar sample volume (500 μL) of calibration-standard solutions, blank water, and untreated feces supernate were prepared by the same LLE and derivatization procedure and analyzed together in the same series using instrumental analysis by LC-MS/MS. Cortisol concentrations in supernate samples were calculated by internal standard calibration.

#### 2.3.3. Extraction

LLE was optimized by the processing of 500 μL supernate samples that had been deconjugated by incubation with 50 μL of ammonium acetate buffer (pH 6) and 20 μL *Helix Pomatia* enzyme solution for 60 min at 60 °C. To these samples, 700 μL of either a water-immiscible solvent (MTBE, ethyl acetate, or butanol) or a water-miscible solvent (ACN or 1-propanol) was added for performing, respectively, either LLE or salting-out-assisted liquid–liquid extraction (SALLE). The saturated NaCl solution was then added, either 100 μL or 600 μL in case of LLE, and 600 μL in case of SALLE. After shaking and centrifugation, 400 μL of the top solvent phase was transferred to an empty vial containing 50 μL of internal standard user solution, followed by evaporation and derivatization, as described above. All experiments were performed in triplicates and allowed optimization of relative extraction recovery measured by the response factor (analyte peak area/internal standard peak area) and simultaneous assessment of the degree of ion suppression as the reduced peak area of internal standard. The optimization aimed for the highest combined effect of extraction recovery and reduced ion suppression to increase the detectability of cortisol.

### 2.4. LC-MS/MS

Derivatized sample extracts were analyzed by using an Acquity UPLC (Waters, Milford, MA, USA) coupled with a Quattro Premier XE mass spectrometer (Waters). A measure of 20 μL of sample volume was injected onto a BEH C18 reversed-phase column with 2.1 mm ID, 100 mm length, and 1.7 Å particle size. The mobile phase consisted of (A) 0.2% (*v*/*v*) concentrated ammonium hydroxide in water mixed with (B) methanol at a flow rate of 0.25 mL/min. The linear step gradient was as follows: 0 min (1% B), 3.5 min (70% B), 4–4.5 min (95% B), 4.6–6 min (1% B).

Hydrazone derivatization products of cortisol (cortisol–4-ABH) and D4-labeled cortisol internal standard (D4 cortisol–4-ABH) were ionized by positive electrospray ionization (ESI+) using 1 kV capillary voltage and 50 V cone voltage. Detection was performed by multiple-reaction monitoring (MRM) using 35 V collision voltage. MRM transitions were with the following *m*/*z* values: 496.25 > 119.8 for cortisol–4-ABH and 500.25 > 119.8 for D4 cortisol–4-ABH. The hydrazones were eluted with a retention time of 4.2 min when using the UPLC conditions described above.

### 2.5. Analytical Performance Characteristics

#### 2.5.1. Final Method

The method for measuring the total cortisol in fish feces was established as follows:Determine exact weight of 0.2–0.5 g fish feces in a microcentrifuge vial.Add 1000 µL water to the vial, followed by 1 min shaking and 5 min centrifugation at 2000× *g*. Total supernate volume is estimated by adding water and feces volumes together, assuming a feces density of 1 g/mL.Transfer 500 µL particle-free supernate to a new vial, then add 50 µL ammonium acetate buffer and 20 µL enzyme solution, before incubation at 60 °C for 60 min. Either beta-glucuronidase enzyme from *Helix pomatia* or from abalone snails (Abalonase™) was applied in the final method.Performed LLE procedure by adding 50 µL internal standard user solution, 100 µL saturated NaCl solution, and 700 µL MTBE to the vial. This step was also performed with 500 µL of calibration-standard solutions, blanks, and a quality control sample (pooled feces supernate sample).Shake the vial for 1 min, followed by centrifugation at 2000× *g* for 5 min.Transfer 400 µL of top solvent phase to a new vial, followed by evaporation to dryness in a vacuum centrifuge at 60 °C for 20 min.Add 20 µL of derivatization reagent solution to the vial, followed by 30 min incubation at room temperature, and finally add 40 µL water.Transfer the derivatized sample extract to an autosampler vial with microvolume insert (0.3 mL) for LC-MS/MS analysis and quantification of cortisol in feces supernate by internal standard calibration.The concentration of cortisol in fish feces should be reported in *w*/*w* units of ng/g (cortisol/feces), which we calculated as follows:
(1)$$\frac{\left[\mathrm{concentration}\;\mathrm{in}\;\mathrm{feces}\;\mathrm{supernate}\right]\times \left[\mathrm{total}\;\mathrm{supernate}\;\mathrm{volume}\right]}{\left[\mathrm{weight}\;\mathrm{of}\;\mathrm{feces}\right]}$$

#### 2.5.2. Procedure

Repeatability (intra series precision) and reproducibility (intra series precision) were investigated by analyzing 10 replicates of the pooled feces supernate the same day and between days, and with a new calibration-standard curve prepared each day. Limit of detection (LOD) and limit of quantification (LOQ) were investigated by analyzing 10 replicates of a surrogate blank sample (water) and by calculating concentrations at response factors made up of average blank plus either 3 times the standard deviation (SD) or 10 × SD of the blank for estimating the LOD and LOQ, respectively. Linearity was assessed from the calibration curve of calibration-standard solutions extracted together with samples. Accuracy was assessed by analyzing pooled feces supernate, both unspiked and spiked to three concentration levels with known amounts of cortisol.

## 3. Results

### 3.1. Method Development

#### 3.1.1. Instrumental Analysis

Cortisol and D4 cortisol reacted with 4-ABH in methanol at room temperature to produce a high yield of hydrazones after 30 min (Figure 1A). Peak splitting was observed in the chromatogram when injecting 20 µL of derivatized cortisol in pure methanol, while dilution with water to 33% methanol resulted in a sharp peak. About 4.5 times higher peak height was observed for derivatized cortisol when using 0.2% ammonium hydroxide (pH 11) in the mobile phase instead of the more conventional 0.2% formic acid (pH 2.5). The high-pH mobile phase was compatible with the BEH C18 stationary phase and was applied in further work towards the final method. Protonated molecular ion adduct of derivatized cortisol was the dominating signal in the mass spectrum and was therefore applied for MRM detection. The observed fragment ions of protonated hydrazones of both cortisol and D4 cortisol had *m*/*z* 119.8 and were assigned to the common oxonium ion derived from the reagent part of the hydrazone molecules (Figure 1B). With optimized conditions, the peak height of derivatized cortisol was 10 times higher than the peak height of the same amount of underivatized cortisol.

#### 3.1.2. Maximizing Deconjugation

Chemical hydrolyses with both acid (H_2_SO_4_) and base (NH_3_) all resulted in lower measured cortisol concentrations than untreated feces supernate (Figure 2A). Deconjugation with beta-glucuronidase from *Helix pomatia* clearly released more cortisol from feces supernate than the three other enzyme solutions tested (Figure 2A) and was more efficient at pH 6 vs. pH 5 (Figure 2B). Adding 20 µL of this enzyme solution was found to be sufficient, and an increase above this volume was not found to release significantly higher amounts of cortisol (Figure 2C). Similarly, incubation at 60 °C for more than 1 h did not produce significantly higher concentrations of cortisol in the fish feces supernate samples (Figure 2D).

#### 3.1.3. Optimization of Extraction

In the salting-out-assisted liquid–liquid extraction (SALLE) experiments, phase separation was achieved for acetonitrile and propanol when adding as much as 600 µL of saturated NaCl to the supernatants. SALLE with acetonitrile resulted in the highest relative recovery of cortisol (Figure 3A). However, SALLE with either acetonitrile or propanol both resulted in severe ion suppression for cortisol, caused by coextracted matrix components (Figure 3B). In the liquid–liquid extraction (LLE) experiments, comparable extraction recoveries were observed for MTBE, butanol, and ethyl acetate. For MTBE, a small but significant increase in recovery was observed when adding more salt (600 µL vs. 100 µL of saturated NaCl). When extracting with MTBE, however, the addition of more salt also resulted in largely more ion suppression (Figure 3B). By assessing the combined effect of recovery and ion suppression, the highest overall detectability of cortisol was achieved by LLE with MTBE and with 100 µL of saturated NaCl added.

### 3.2. Analytical Performance Characteristics

Method LOD of 0.04 ng/mL and LOQ of 0.09 ng/mL were estimated for part of the method involving analysis of 500 µL supernate samples. By considering a 3.8 times dilution of wet feces with water, this corresponded to a total method LOD of 0.15 ng/g and LOQ of 0.34 ng/g in the original samples. Later inspection of the signal-to-noise (S/N) ratio in chromatograms, from analysis of individual fish feces deconjugated with Abalonase™ enzyme, found to contain cortisol around 1 ng/g, did in fact confirm such a low LOQ. The calibration curve was found to be linear (R^2^ > 0.997) up to 100 ng/mL, corresponding to a concentration in fish feces of 380 ng/g. Accuracy was investigated by spiking cortisol into aliquots of pooled feces supernate, which resulted in the following sample concentration increases: 4.8, 9.6, and 19 ng/mL. By analyzing both spiked and unspiked samples, the recovery estimated at these three levels were 114 ± 12, 126 ± 16, and 127 ± 6%. By analyzing the very homogenous pooled feces supernate, both intra-series and intermediate precision had a CV of 11%. However, by analyzing individually weighed fish feces samples according to the final method protocol, a somewhat higher CV of 17% was estimated. The chromatograms for the analysis of a fish feces supernate sample containing 1.50 ± 0.19 ng/mL are shown in Figure 4. Carryover was estimated to be 0.05 ± 0.02% (*n* = 6) in blank injections directly after the highest calibration standard.

### 3.3. Application of the Method

The novel method was applied for analysis of feces collected from individual fish (Atlantic salmon) a few days after transfer to seawater. For this work, we applied the highly characterized commercial enzyme from Abalone snails (Abalonase™) which was a pure beta-glucuronidase. High levels of cortisol, 437 ± 293 ng/g, were found four days after transfer but the levels gradually decreased to 74 ± 55 ng/g at day eight after transfer (Figure 5).

## 4. Discussion

Because of the rather low concentrations of cortisol in fish feces, at the part-per-billion (ppb) level, it soon became evident that an improved detectability method was needed. This was partially achieved by the derivatization with 4-ABH and optimization of LC-MS/MS conditions, especially by using high injection volume and ammonium hydroxide in the mobile phase. When we previously investigated several derivatization reagents for improved detectability of steroid hormones in saliva, the hydrazones formed with 4-ABH were found to produce the highest increase in signal response for cortisol, and again four times higher when using ammonium hydroxide in the mobile phase [20]. Protonation of amines by the ammonium ion (NH_4_^+^) in positive electrospray ionization (ESI+) was described in an early work by Espada and Rivera-Sagredo [21]. However, we previously discovered that many of the hydrazone derivatization products exhibited chromatographic peak splitting, which was the case also for cortisol. This phenomenon was not fully understood at the time and led to some erroneous speculation over possible isomerization. In the present work, we discovered that a dilution of samples with water, after derivatization, reduced the sample solvent elution strength and resulted in the elimination of the peak splitting. The final sample solvent volume was kept low (60 µL) and contained only 33% methanol added from the derivatization solution.

An additional contribution to improved detectability was the use of an optimized LLE in the sample preparation. However, fish feces contain many polar matrix components that will be coextracted in various amounts together with cortisol. Indeed, increased ion suppression was observed when using polar solvents such as butanol, acetonitrile, and propanol for the extraction. Fortunately, the rather nonpolar MTBE was found to extract a satisfactory amount of cortisol from the supernate. Interestingly, attempts to increase extraction recovery by adding more salt also produced more ion suppression. This was explained by the fact that more salt makes the aqueous phase even more polar and will thereby transfer more matrix components to the solvent phase.

Deconjugation of cortisol phase II metabolites was best achieved by using beta-glucuronidase from *Helix pomatia*. The other three enzymes all resulted in the suboptimal release of cortisol from the feces supernate samples. The most straightforward explanation for this is that cortisol is mainly conjugated with sulfate, and the fact that the solution from *Helix pomatia* contains some amount of sulfatase: <7500 U/mL according to specifications. The activity of the contained sulfatases could be especially efficient for deconjugation of cortisol, explaining the large difference between this enzyme solution and a commercial product with a more controlled, although slightly lower, sulfatase activity (Abalonase™ +). For investigation of stress in individual fish, we finally decided to apply the highly characterized pure beta-glucuronidase from the abalone snail (Abalonase™). Although this enzyme was only the second most efficient for deconjugation, it was still believed to offer a more consistent release of cortisol. It was assumed to be less affected by lot-to-lot variations, which are more likely encountered with the enzyme solution from *Helix pomatia* that was a crude gastric juice solution from the snails.

This work is the first to present a method for quantification of either free or conjugated total cortisol in fish feces by LC-MS/MS. We utilized the Quattro Premier XE (Waters) mass spectrometer which is more than 10 years old and significantly less sensitive than recent instrument models. Still, we were able to achieve a satisfactory detectability for cortisol at low concentration levels. This can be attributed to the combined effect of LLE enrichment, derivatization, increased injection volume, and by using ammonium hydroxide instead of formic acid in the mobile phase. Samples from individual fish were analyzed, thereby demonstrating the method as a new tool for monitoring stress in farmed fish. In fish recently transferred from freshwater to seawater, the level of stress is known to be high [22]. Indeed, our new method was able to detect significant stress in fish on day four after transferal to the sea. The high standard deviation for these samples indicated a high variation in the stress level among individual fish directly after release. Moreover, the mean cortisol level decreased steadily from above 400 ng/g to below 100 ng/g on day eight after transfer to seawater. Hence, the method shows great potential for future monitoring of fish welfare. More work is needed to establish reference values, but preliminary data indicate that the lower levels of cortisol found on day eight after transfer were still significantly elevated compared to levels found in unstressed fish. The new method presented here could also be useful in more fundamental studies of fish endocrinology, for instance in studies of cortisol coregulation [23].

A previous study with parrotfish [17], where fecal cortisol was quantified by HPLC-UV, found a baseline cortisol concentration of 3.4 ± 1.1 ng/g, increasing to 36.9 ± 7.3 ng/g in stressed fish. These levels are in line with our findings; although, our data reveals that the sum of free and conjugated cortisol in feces can reach levels of more than 1000 ng/g in feces from Atlantic salmon under stressful conditions. Future work may involve testing other enzymes for consistent deconjugation of both glucuronidated and sulfated cortisol. It will also be necessary to fully explore the biomarker potential of the method by analyzing samples collected from fish after other stressful events, for instance during an outbreak of disease.

## 5. Conclusions

To our knowledge, this paper is the first to present a highly optimized sample preparation method for quantification of cortisol in fish feces by LC-MS/MS. The method was applied and found suitable for stress monitoring in farmed Atlantic salmon. It allowed us to conclude that fish released into seawater experienced this as a stressful event, but also that the level of stress was decreased significantly after eight days in the sea. We strongly believe that the method will be useful in future studies on fish welfare and perhaps also in more fundamental studies of fish endocrinology.

## Figures and Tables

**Figure 1 molecules-27-02481-f001:**
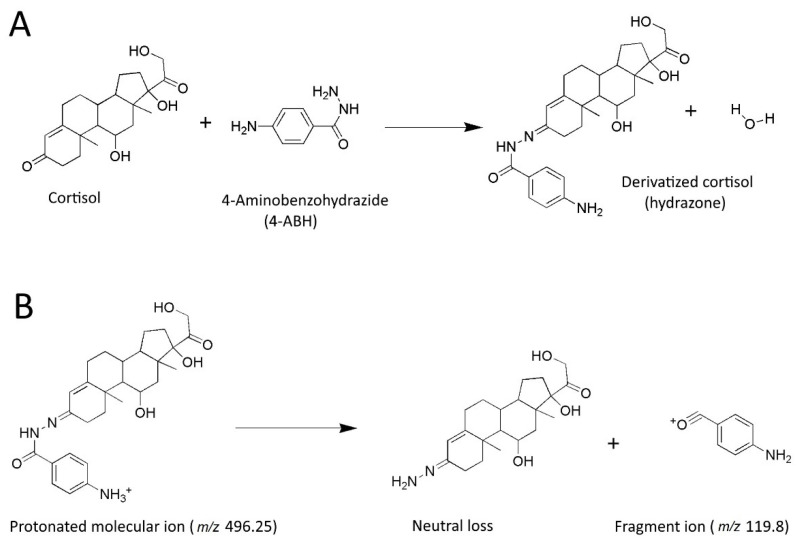
Molecular structures for illustration of the following: (**A**) derivatization reaction between cortisol and 4-ABH (10 mg/mL) to the hydrazone; (**B**) collision-induced dissociation (CID) reaction taking place in the collision cell of the mass spectrometer for fragmentation of the protonated molecular ion of derivatized cortisol (capillary voltage: 1 kV and cone voltage: 50 V) and formation of the oxonium fragment ion (collision energy: 35 eV).

**Figure 2 molecules-27-02481-f002:**
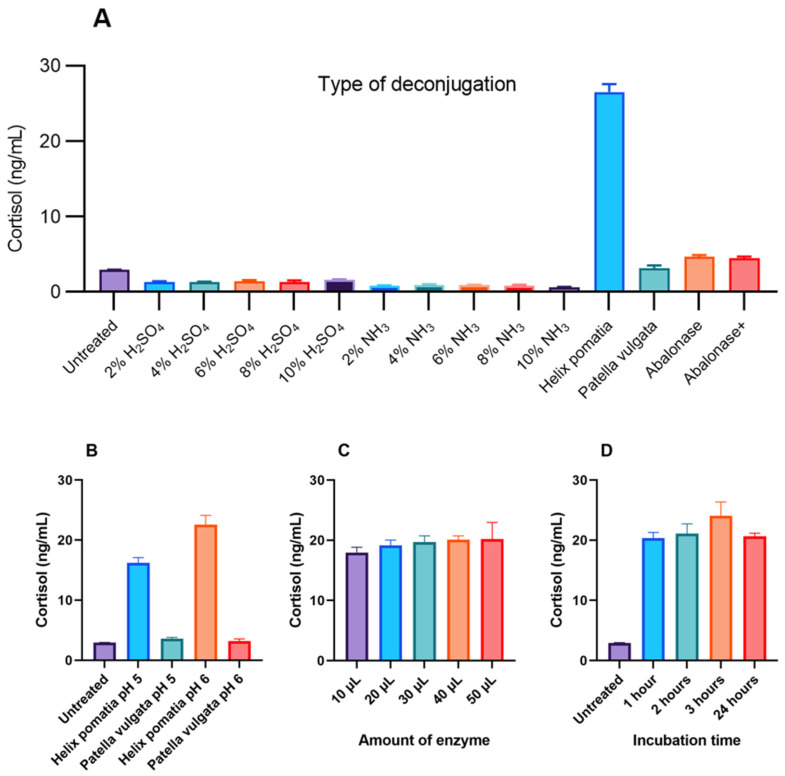
Deconjugation experiments showing various amounts of cortisol released from the fish feces supernate, measured as mean cortisol concentrations with 1 × SD error bars. (**A**) Acidic (H_2_SO_4_) and basic (NH_3_) hydrolysis compared with enzymatic deconjugation using four different beta-glucuronidases; (**B**) deconjugation with two different enzyme solutions tested at pH 5 and 6; (**C**) deconjugation at pH 6 with different volumes of *Helix pomatia* enzyme solution added; (**D**) deconjugation at pH 6 with 20 µL of *Helix pomatia* enzyme solution added and different times of incubation at 60 °C.

**Figure 3 molecules-27-02481-f003:**
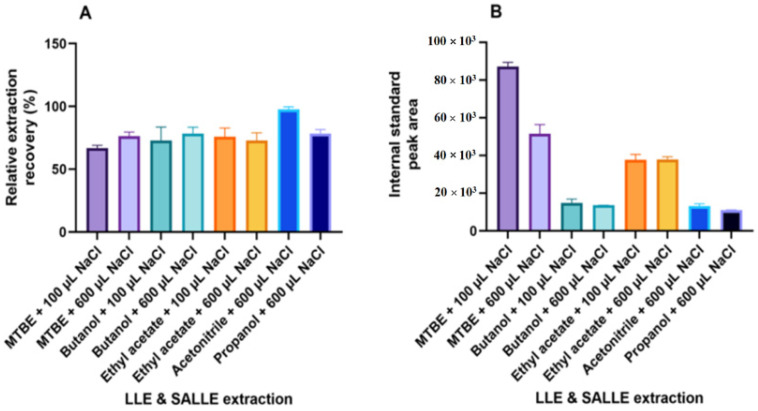
Liquid–liquid extraction (LLE) and salting-out-assisted liquid–liquid extraction (SALLE) experiments showing (**A**) relative extraction recoveries with different solvents and salt added and (**B**) peak area of the internal standard added after extraction to assess the degree of ion suppression caused by coextracted matrix components.

**Figure 4 molecules-27-02481-f004:**
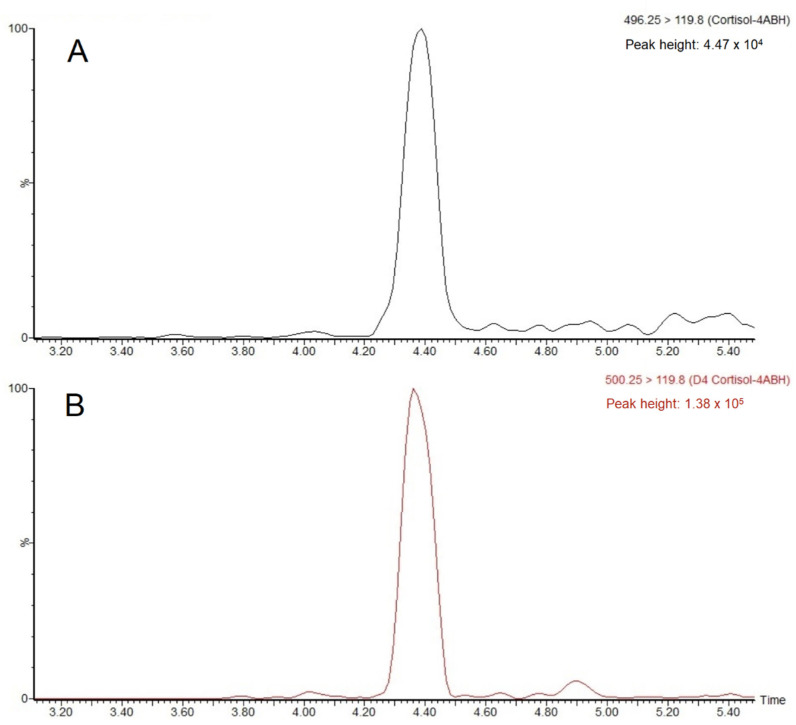
MRM-chromatograms of (**A**) 4-ABH-derivatized cortisol from fish feces supernate sample containing 1.50 ± 0.19 ng/mL of cortisol and (**B**) 4-ABH-derivatized D4 cortisol internal standard.

**Figure 5 molecules-27-02481-f005:**
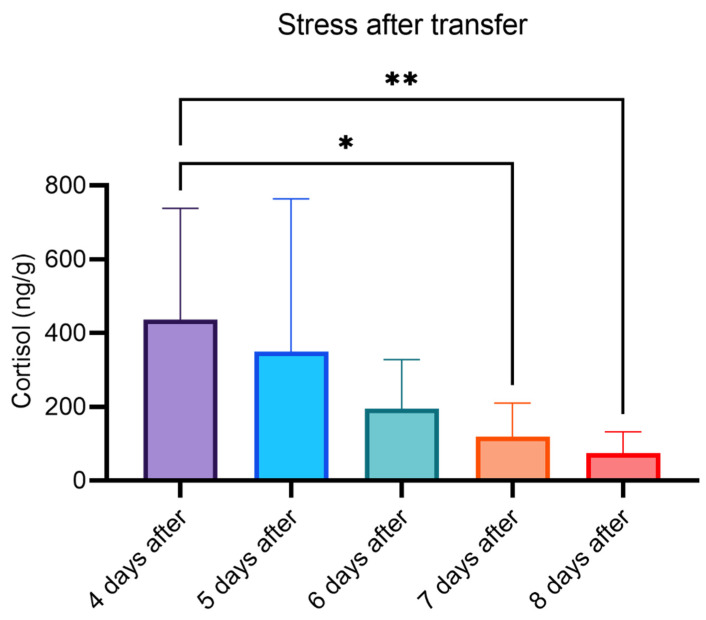
Concentrations of cortisol measured in samples of feces from individual fish transferred from freshwater to seawater. The mean cortisol concentrations were: 437 ± 302 (*n* = 18), 349 ± 414 (*n* = 10), 195 ± 133 (*n* = 10), 119 ± 92 (*n* = 9), and 74 ± 58 ng/g (*n* = 10), after 4, 5, 6, 7, and 8 days, respectively. In comparison with 4 days after, the mean levels were significantly lower 7 days after (* *p* = 0.027) and 8 days after (** *p* = 0.0057).

## Data Availability

Not applicable.

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
