# Peer review of "Monitoring Farmed Fish Welfare by Measurement of Cortisol as a Stress Marker in Fish Feces by Liquid Chromatography Coupled with Tandem Mass Spectrometry"

_molecules, 2022, doi:10.3390/molecules27082481_

Round 1

Reviewer 1 Report

The article is an useful contribution to analytical methodologies for monitoring of biological markers in order to assess the farmed welfare of animals. In this case, it is about cortisol level in farmed fish. The article is in general well presented. However there are some issues, in my opinion, which has to be corrected:

  1. Poor bibliographic research – even if the theme is a niche one, there are many other scientific contributions regarding the correlation of cortisol levels and stress in animals.
  2. For each mean value, such as average weight of fish, the N values and SD or RSD% have to be indicated.
  3. Some methodologies are too in details presented, such as in 2.1 section. Some are obvious steps in an analytical laboratories. However it allows the reader to reproduce exactly the experiment. But in my opinion the description from rows 74 to 83, on page 2, has to resume just to present the concentrations and solvents for each type o solutions, stock and working.
  4. Througout of article use “Method analytical performances” instead of “validation”– a method validation means much more than it was presented.
  5. In the section < Results>, move all discussions and interpretation in the <Discussion> section or combine both in one new section <3. Results and discussions>.
  6. The Discussion section has to be further improved in connection to: other LC-MS/MS methods for cortisol monitoring (particularities, improvements of the present method); interpretation of the behaviour of the cortisol profile (decreasing of the cortisol levels in time) (i.e. Fürtbauer, I., Heistermann, M. Cortisol coregulation in fish. Sci Rep6, 30334 (2016). https://doi.org/10.1038/srep30334 )
  7. The <Conclusions> section is missing – it is a mandatory section for any kind of scientific article.

As a conclusion, after some changes mentioned above and further improvement of the bibliographic research, the article could be published in the journal Molecules.

Author Response

  1. Added new text (line 33 and lines 36-43) and five new references as requested by Reviewer 4.
    Added new text (line 47-52): “The major reaction control system for stress in animals is the hypothalamic–pituitary–adrenal (HPA) axis [8]. An equivalent to this system is the hypothalamic–pituitary–intrarenal (HPI) axis found in teleost fish [9,10]. Triggering of the system during stressful events elevates the production of corticosteroid hormones, resulting in higher levels of cortisol measured in the blood [11–13]. It is therefore possible to study stress in fish by monitoring the level of cortisol”
    Added new references: Miller 2018, Bernier and Peter 2001, Pijanowski 2015, New et.al 1996, Kartashova et.al. 2021, Champagne et.al. 2018.
    Re-written text in line 54-57 to now read: “However, sampling of blood from fish is stressful by itself and may lead to erroneous results. Measurement of cortisol in feces is an alternative approach and could potentially correlate with stress experienced some time before the sampling is done.”
  2. Corrected with mean average +/- SD and numbers for fish samples on line 105.
    Figure 4 has been updated with significance indicators and new figure text high-lightening details on mean values with SD, as well as N.
  3. If possible, we would like to keep the experimental details for transparency and reproducibility.
  4. “Method validation” replaced with “analytical performance characteristics” in chapter 2.3.1 and in headlines of chapter 2.5 and 3.2. Replaced “Method Validation Procedure” with “Procedure” in headline of chapter 2.5.2.
  5. In the Result chapter we found some interpretation now in line 286 (“but still acceptable”). This text was deleted. After this, we believe the Result chapter is strictly presenting observations.
  6. We could not find other papers describing LC-MS/MS for cortisol in fish feces. However, we agree that we need some final discussion on the new method, so we added the text now in lines 360-365: “We utilized the Quattro Premier XE mass spectrometer which is more than 10 years old and significantly less sensitive than recent instrument models. Still, we were able to achieve a satisfactory detectability for cortisol at low concentration levels. This can be at-tributed to the combined effect of LLE enrichment, derivatization, increased injection volume, and by using ammonium hydroxide instead of formic acid in the mobile phase.”
    We thank the reviewer for informing us about the paper by Fürtbauer et.al. and this is clearly a paper describing a more fundamental endocrinology study. Unfortunately, we have no data to discuss in relation to their study, but have now included the reference and text now at line 375: “The new method presented here could also be useful in more fundamental studies of fish endocrinology, for instance in studies of cortisol coregulation [23]”

  7. Added chapter 5. Conclusion with the following text: “To our knowledge, this paper is the first to present a highly optimized sample preparation method for quantification of cortisol in fish feces by LC-MS/MS. The method was applied and found suitable for stress monitoring in farmed Atlantic salmon. It allowed us to conclude that fish released into seawater experienced this as a stressful event, but also that the level of stress was decreased significantly after eight days in the sea. We strongly believe that the method will be useful in future studies on fish welfare and perhaps also in more fundamental studies of fish endocrinology.”

Reviewer 2 Report

The paper entitled “Monitoring Farmed Fish Welfare by Measurement of Cortisol as Stress Marker in Fish Feces by Liquid Chromatography Coupled with Tandem Mass Spectrometry” authored by Vanessa Andrea Meling, Kjetil Berge, David Lausten Knudsen, Per Ola Rønning and Cato Brede, presents a novel method for measurement of cortisol in fish feces based on enzymatic hydrolysis, liquid-liquid extraction, derivatization and liquid chromatography coupled with tandem mass spectrometry analysis. The paper is nicely written, well organized and should be accepted for publication after the points below are addressed.

  1. Too few references are cited in the paper. The authors should cite more researches related to the topic, particularly in the Introduction section.
  2. In the abstract, in the sentence “Method repeatability and intermediate precision were acceptable, both with a CV of 11%.” the abbreviation CV should be defined.
  3. It would be useful if the results are presented in ppm, ppb or ppt units as well.
  4. Latin names should be written in italic (for example Helix pomatia)

Author Response

  1. This issue was also mentioned by two other reviewers, so we have added more references.
  2. Added “coefficient of variation (CV)” in line 26
  3. Added new text now on line 319: “Because of the rather low concentrations of cortisol in fish feces, at the part-per-billion (ppb) level, it soon became evident that an improved detectability was needed.”
  4. Corrected to italic for latin names.

Reviewer 3 Report

The research entitled “Monitoring Farmed Fish Welfare by Measurement of Cortisol as Stress Marker in Fish Feces by Liquid Chromatography Coupled with Tandem Mass Spectrometry” by Meling et al deals with development of new method for analysis of cortisol in fish feces by LC-MS/MS analysis. The authors demonstrated the applicability of their method for a concentration range of 0.34-389 ng/g, which is remarkable. It is an interesting manuscript could be considered for possible publication in molecules after revisions.

Abstract: Sufficient. Expand CV: coefficient of variance

Introduction: line 48 and 57, abbreviate only once (LC-MS/MS).

Section 2.3.3: Expand SALLE.

Figure 1: m/z should be italics (throughout) and provide the conditions for each step either in scheme or in figure legend. Do authors have any information about ionization of 4-ABH derivatized cortisol in negative mode? Why they choose positive mode. The sensitivity comparisons should be described.

-Provide the MS, MS/MS and EIC of the cortisol and its labelled form as a new figure.

Line 213:214: “With optimized conditions, the peak height of derivatized cortisol was 10 times higher than the peak height of the same amount of underivatized cortisol”. Provide the experimental data as supplementary figure to prove derivatization improved sensitivity by 10 folds.

Section 3.2: Method validation data should be provided as a table. Recoveries should be represented with standard error. The authors didn’t show any results on matrix effect. As this parameter is highly important, the authors must address it in the revised manuscript

Figure 4: Why the 4 day and 5 day samples have high standard deviation. Almost 2 folds. The parameters effect these results should be discussed. Also, for all figures in the legend provide the “n” for each group and is it mean ±SD or SE.

To say their results are significantly different (especially Fig 4) authors must apply appropriate statistical test (one-way ANNOVA or t-test) and provide the information as a separate section. Line 284-286: Use of ammonium hydroxide as mobile phase increase the signal intensity by 4 folds. I think in this kind of situation, carryover experiments should be performed. The authors must include the percentage carry over. The authors may refer the following study (PMID: 29740670) employed alkali mobile phase to improve the sensitivity. Also, the discussion on previous method of cortisol analysis and their limitations should be addressed. The applicability of un-targeted LC/MS techniques in fish sample analysis are lacking (PMID: 32414047, PMID: 34053529).  I suggest including in the revised MS.

Author Response

  1. Added “coefficient of variation (CV)” in line 26
  2. Abbreviation only once, deleted text now on line 71.
  3. Now in line 149, expanded SALLE with “salting-out assisted liquid-liquid extraction”
  4. m/z in italic throughout and in Figure 1, updated Figure 1 legend with conditions for each step.
  5. Ionization in negative mode was briefly investigated for both cortisol and 4-ABH derivatized cortisol. In both cases, strong signal in positive mode but not in negative mode. We would have mentioned it if negative ionization could be an alternative, but in this case no alternative.
  6. We think it is most relevant to provide the MRM-chromatograms for derivatized cortisol and internal standard in a real sample. This is now Figure 4.
  7. We have already investigated the difference in signal response between derivatized and non-derivatized cortisol, for many reagents and different mobile phases, and published these results in a previous paper, please see reference 20. We clearly mention this in the paper and therefore think this topic is covered sufficiently in the beginning of the Discussion.
  8. Section 3.2. We think that only one analyte may not warrant a separate table for presenting validation results, so we prefer not to include a new table.
    We have updated the reported recoveries with standard deviations.

    The matrix effects (ion suppression) are presented in Figure 3B and are comprehensively discussed in the text. Perhaps the reviewer is not aware of the possibility of this type of investigation. We present a rather elegant double experiment, involving the addition of same amount of internal standard to all extraction experiment after the extraction. Hence, allowing for investigation of both relative extraction recovery, and ion suppression (matrix effects) in the same experiment. We think this was properly explained.

Figure 4 (now Figure 5) has been updated to indicate significant differences between the days with the p-value in the figure legend.
Figure legend is also updated with standard deviation and number of samples in each group. We added some discussion on this in line 369: “The high standard deviation for these samples indicated a high variation in the stress level among individual fish directly after release.”

Added new text with info on carry-over in line 288: “Carry-over was estimated to be 0.05 +0.02% (n=6) in blank injections directly after the highest calibration standard.”

We are grateful for the suggested reference on alkalic mobile phase, but this is for sphingolipids. In fact, we are using 0.1% ammonia and pH 11 for phosphatidylethanol LC-MS/MS analysis at Stavanger University Hospital, for negative electrospray ionization.
However, for amines such as for our derivatization reagent (4-ABH) functional group it is not so common to use ammonium hydroxide and high pH in the mobile phase, and most people use 0.1% formic acid. The ammonium ion (NH4+) can act as proton donor for analytes with amine functionality, such as 4-ABH derivative. Hence, the more efficient ESI+ ionization for this analyte. Please see our paper reference 20 where we have covered this for 4-ABH cortisol.

To further explain, we added a very early paper describing this phenomenon and new text on line 326: “Protonation of amines by the ammonium ion (NH4+) in positive electrospray ionization (ESI+) was described in an early work by Espada and Rivera-Sagredo [21].”

We would like to keep a strict focus on cortisol measurement in our paper, and not discuss fish sample analysis so much in general. Hope for understanding. We could have added many  more references on hormone analysis by LC-MS/MS, but this is very much covered by the excellent review with reference number 18.

Reviewer 4 Report

Manuscript ID: Molecules-1638154

Title: Monitoring Farmed Fish Welfare by Measurement of Cortisol as Stress Marker in Fish Feces by Liquid Chromatography Coupled with Tandem Mass Spectrometry

xxxxxxxxxxxxxxxxxxxxxxxxxxxxxxxxxxxxxxxxxxxxxxxxxxxxxxxxxxxxxxxxx

  1. The same article available online, can be viewed through the following link. Authors should explain how the communicated article is simultaneously available online.

https://www.preprints.org/manuscript/202203.0012/v1

  1. Reference 1 needs to be updated, the article or the data may be around 3-4 years old.
  2. Introduction line 34. Authors should explain, the industries face what kind of health and welfare issues related to fishes. Introduction needs to be elaborated, should be updated with the similar works in recent days.
  3. How the sampling of cortisol from feces differ from the that in blood, did author compared the two sampling procedure to justify their claims of erroneous results.
  4. Line 74, what is intermediate solution, if it is the dilution of the stock then why it was not diluted in water as in case of the calibration samples.
  5. Why deconjugation and derivatization experiment was required, MS did not detect the analyte?
  6. The conjugation experiments involve the elevated temperature (60°C), acid and base.........did author checked the stability of the target analyte? Authors should refer to few article dealing in the stability of cortisol at different conditions involved in the study.
  7. Did author developed new method or have utilized reported method, detailed validation report is required.
  8. Recovery estimated in section 3.2 at three points were 114, 126 and 127 %, what is the permissible range for recovery in such extractions.?
  9. Few article suggest the detection of cortisol in fertilized and unfertilized fish eggs, did authors try fish egg to evaluate cortisol level. It will be informative if author compare the three matrix eggs, feces and blood.

Author Response

  1. The article was automatically uploaded by the publisher to the preprint server, as part of the submission process to Molecules. During electronic submission to Molecules, we were asked by the publisher if this was accepted, and we answered “yes”.
  2. Added a more recent reference (Rocha et.al. 2022) and updated the manuscript with the following text on line 33: “and is now producing more than half of the total volume in Europe.”
  3. Added sentences on line 36-43: “With the typical production in surface-based cages, fish health can be challenged by a range of events, including weather, parasites, algae, jellyfish, diseases, reduced oxygen, and crowding by net deformation [4]. Following a jellyfish bloom occurrence for example, both direct and indirect effects on fish health can be observed. Jellyfish is known to cause biofouling of nets, which reduces water flow and leads to accumulation of solids and an oxygen reduction [5]. Furthermore, fish can be exposed to strong currents and waves when moving nets to new locations [6]. Stressful events like these directly reduce the appetite and growth rate of fish [7] and make them more susceptibility to diseases [5].”
    Added references: Sievers et.al. 2022, Hvas et.al. 2021, Clinton et.al. 2021, and Sambraus et.al. 2018.
  4. Unfortunately, we do not have samples of blood from the same fish. We agree that we have made some premature claims, so we have corrected the text in line 54-57 to now read: “However, sampling of blood from fish is stressful by itself and may lead to erroneous results. Measurement of cortisol in feces is an alternative approach and could potentially correlate with stress experienced some time before the sampling is done.”
  5. Stock solution of cortisol is made by weighing and is of very high concentration. Intermediate solution is more diluted but still rather high concentration compared to calibration standards. Added new text on line 92: “Solutions prepared in methanol are stable at 4 ⁰C for at least 2 years in our experience (Stavanger University Hospital).”
  6. This need for derivatization is due to very low levels in feces. Naturally, in some samples, deconjugation may not be necessary, but this is typical in many cases of trace analysis.
    We think this is already well covered in the Discussion chapter.
  7. There are many publications on LC-MS/MS measurement of steroid hormones in urine utilizing beta-glucuronidase at 50-60 ⁰C for up to 4h. Therefore, we think the issue of stability has been covered by others and can be omitted in our paper.
  8. This is a new method, and no validation report is available. We have therefore replaced “Method validation” with “analytical performance characteristics” to make it clear that we investigated these parameters more briefly.
  9. Since we have no other laboratory to compare results with, we needed to investigate method accuracy by recovery experiments. Usually, these experiments are not perfect, but typically recovery in the range 75-125 % is considered acceptable. Our average recovery is within this range. In the future, we would like to do more investigations of method accuracy, but so far we are the only laboratory to do this by LC-MS/MS, and no reference material is available.
  10. Thank you for this interesting suggestion. Unfortunately, we do not have fish egg samples at this moment. However, we have plans to include a comparison between blood and feces in future follow-up work. Perhaps fish eggs could also be sampled.

Round 2

Reviewer 3 Report

The authors had improved their manuscript and answered most of my comments. It is now suitable for publication in molecules.

Author Response

Thank you for your comments to improve our paper.

Reviewer 4 Report

Line 52-53.  However, sampling of blood from fish is stressful by 52 itself and may lead to erroneous results..... This line should either be removed or should add any reference.

Point 8 in the authors reply “more briefly”, what does it mean?

Author Response

Thank you for your comments to improve our paper.

  1. We agree that this statement is still speculative, so we have removed the sentence.
  2. As suggested by another reviewer we replaced "method validation" with "analytical perfomance characteristics" which are those results presented in the paper. It is perhaps a less thorough (brief) investigation than a full method validation, but sufficient information about new methods to show their potential.